# Trends in Biological Ammonia Production

**DOI:** 10.3390/biotech12020041

**Published:** 2023-05-19

**Authors:** Adewale Adeniyi, Ibrahim Bello, Taofeek Mukaila, Niloy Chandra Sarker, Ademola Hammed

**Affiliations:** 1Environmental and Conservation Sciences, North Dakota State University, Fargo, ND 58102, USA; 2Agricultural and Biosystems Engineering, North Dakota State University, Fargo, ND 58102, USA

**Keywords:** biological ammonia, bioprocessing, bioengineering, fermentation, enzyme immobilization

## Abstract

Food production heavily depends on ammonia-containing fertilizers to improve crop yield and profitability. However, ammonia production is challenged by huge energy demands and the release of ~2% of global CO_2_. To mitigate this challenge, many research efforts have been made to develop bioprocessing technologies to make biological ammonia. This review presents three different biological approaches that drive the biochemical mechanisms to convert nitrogen gas, bioresources, or waste to bio-ammonia. The use of advanced technologies—enzyme immobilization and microbial bioengineering—enhanced bio-ammonia production. This review also highlighted some challenges and research gaps that require researchers’ attention for bio-ammonia to be industrially pragmatic.

## 1. Introduction

Ammonia, a compound with the chemical formula NH_3_, is composed of two of the most ubiquitous elements on Earth—nitrogen and hydrogen [1,2,3]. Ammonia is colorless and characterized by its pungent odor. Ammonia has a wide range of industrial and agricultural applications due to its unique properties. In nature, ammonia exists in the soil and the environment as a product of ammonia-producing bacteria, plants, animals, and waste decomposition. In its pure form, ammonia is anhydrous and hygroscopic, as it readily absorbs moisture. Anhydrous ammonia can be a gas, a liquid, or a solid, depending on its temperature and pressure. In gaseous form, ammonia is less dense than air and its liquid is less dense than water at standard atmospheric temperature and pressure. Ammonia vapor diffuses readily in the air, and liquid ammonia is soluble in water with a simultaneous release of heat. When dissolved in water, ammonia gas forms ammonium hydroxide (NH_4_OH), a weak base and caustic solution. Under pressure, ammonia gas is easily compressed and forms a clear liquid. Generally, ammonia is corrosive and has alkaline properties. Several other physicochemical properties of ammonia have been published [4,5].

Ammonia was first produced on a large scale in 1913, following the evolution of an artificial nitrogen fixation process developed by German chemists Fritz Haber and Carl Bosch in 1909 [6]. This process, widely known as the Haber-Bosch process, is the reaction of hydrogen gas with nitrogen gas in the presence of an iron catalyst to produce ammonia [7], (Figure 1). At the inception of the Haber–Bosch process, about 20 tons of ammonia was produced per day. In 2018, about 230 million tons of anhydrous ammonia was produced, and it is projected that nearly 290 million metric tons of anhydrous ammonia will be produced in the year 2030 [8].

However, a major challenge with ammonia production is the use of fossil fuels such as natural gas, coal, and oil as feedstock. Concerns have been growing about the sustainability of ammonia production, and more importantly, the effects of its energy use on the environment. Additionally, ammonia production is affected by global energy politics. For instance, the increase in ammonia prices is due to scarcity and an increase in production energy costs. The Russia-Ukraine war has not only caused an increase in energy costs and scarcity but also a 25% global reduction in ammonia fertilizer [9,10,11]. These issues drive an increase in farm input costs, thereby causing an increase in global food prices.

The Haber–Bosch process is presently one of the most prominent emitters of greenhouse gases, accounting for about 1.2% of CO_2_ emissions produced globally. The entire ecosystem is being impacted by higher emissions of nitrous oxide (N_2_O), which has now been classified as the third most significant greenhouse gas after CO_2_ and methane [12]. As a result of these challenges, scientists have been prompted to make findings on better alternatives [2].

Nonetheless, even with the numerous advantages of Haber–Bosch process-based ammonia, the need for large-scale production of ammonia from other sources is requisite. Research on green ammonia and organic ammonia has recently been on the rise. Green ammonia is a term used to describe ammonia produced to accomplish a zero emission target. It is a product of a renewable and carbon-free process, thus effective in the reduction of greenhouse emissions. However, the level of ammonia production was found to be 50,000 tons/year [13], which is considerably low.

## 2. Economic Importance of Ammonia

Ammonia is a highly versatile product that has both domestic and industrial applications. It is commonly used in the production of household cleaning agents for domestic purposes. On an industrial scale, ammonia has a wide range of uses, including use as an extraction solvent, water purification, production of fertilizers, refrigerant gas, plastics, dyes, and explosives [14,15,16,17]. One notable example of ammonia’s industrial usage is in the production of nylon through a chemical process called polymerization [18]. In this process, hexamethylene diamine, which is derived from ammonia, is combined with adipic acid to produce a polymer that can be spun into fibers. These fibers are then used in the production of clothing, carpets, and other textiles. These versatile applications have made ammonia a vital component in global trade and commerce.

### 2.1. Scale of Production

Apart from sulfuric acid, ammonia is the highest-volume chemical commodity produced in the world [3]. East Asia, including China, is the largest producer of ammonia, followed by East Europe and Central Asia, North America, South Asia, and the rest of them (Figure 2). In the United States, the third largest ammonia producer, 14 million MT were recorded in 2021. Nutrien, Koch Industries, and CF Industries are by far the largest producers of ammonia in the United States, with CF Industries alone producing over seven million MT of ammonia in 2021 with its top three production facilities in the United States [19].

### 2.2. Application as Fertilizer

Presently, more than 85% of ammonia is used to produce chemical fertilizer for proper plant development and growth [20]. According to the World Fertilizer Trends and Outlook 2020 report of the Food and Agriculture Organization (FAO), United Nations, the volume of ammonia produced globally is estimated at 150 million metric tons, with a prediction of about 2.3% increment per year [21]. Ammonium sulfate (NH_4_)_2_SO_4_ is the first nitrogenous fertilizer made by BASF, and it was the leading form in which fixed nitrogen was produced before the Second World War. Many forms of ammonia fertilizers are available for different soil types and vegetation. Ammonium nitrate, (NH_4_)NO_3,_ which contains ~35% nitrogen, used to be a leading form of solid ammonia fertilizer compounds. Urea fertilizer, also first manufactured by BASF in 1922, contains more nitrogen (46.6%) and has more advantages over other ammonia containing fertilizers produced prior, making it a leading solid nitrogen fertilizer. In today’s plant agriculture, however, mixed fertilizers are used. They range from combining two or more macronutrients such as potassium nitrate (~13.8% nitrogen) and ammonium phosphates (~10–21% nitrogen) with other granulated materials, usually with selected micronutrients [7].

### 2.3. Fuel Potential

The potential for ammonia to be used directly as fuel to replace fossil fuels in a wide range of applications is also garnering interest. It is anticipated that upon ammonia combustion in engines, the exhaust should be pure nitrogen and water that are safely released into the environment, forming a sustainable and circular cycle that is dubbed the “ammonia economy”. Relative to fossil fuels, ammonia can store chemical energy, which is subsequently released by the breaking and making of chemical bonds. The net energy gain is generated from breaking N-H bonds, which produce nitrogen and water in the presence of oxygen.
4NH_3_ + 3O_2_ ⇌ 2N_2_ + 6H_2_O (1)

The energy storage mechanisms of ammonia are fundamentally the same as those of methane, which has four C-H bonds that can be broken down to release energy. However, in ammonia, where the central atom is nitrogen, nitrogen gas (N_2_) is produced instead of carbon dioxide (CO_2_), which results when methane and other C-H-containing gases are burned. Ammonia can be safely stored in bulk in a liquified form that is achieved by compression to 10 times atmospheric pressure or cooling to −33 °C. In its liquified form, ammonia has an energy density of about 3–3.5 kWh/liter, which is less than that of ethanol and liquified natural gas at 6 kWh/liter but comparable to them [22]. The Mid-West of the US has over 10,000 ammonia storage sites, with the highest densities in Iowa with a storage capacity of ~800,000 tons. Transportation is by multiple means, including pipelines of around 3000 miles that connect 11 states, transporting ~2 MT of ammonia per year [23]. Leakage of these pipes could result in serious health and environmental risks due to the corrosiveness and potential toxicity of ammonia, but it is readily detectable by smell even at very low concentrations that are below levels that could cause any lasting health issues. Nevertheless, stringent controls must be adopted at ammonia storage, transportation, and industrial sites to ensure that the risks of ammonia release are negligible. Apart from its use in nitrogen fertilizer manufacturing and as a zero carbon fuel for many engines, ammonia has also found applications in the explosives, textile, and pharmaceutical industries. However, the energy cost of producing ammonia conventionally might outweigh the energy potential. Therefore, the success of ammonia as a fuel heavily relies on the development of less energy demanding processes, such as biological ammonia routes.

## 3. Ammonia Classification

Ammonia production systems can be classified into three categories based on the carbon emissions from the production processes but not the type of ammonia being produced: brown (or grey) ammonia, blue ammonia, and green ammonia.

### 3.1. Brown (or Grey) Ammonia

The Haber-Bosch process is the conventional method for ammonia production. It is responsible for more than 60% of the ammonia produced globally. Due to its high energy requirement and significant contribution to CO_2_ emissions, the resulting ammonia from the Haber-Bosch process is termed brown ammonia. The Haber-Bosch process is the reaction of nitrogen (N_2_) and hydrogen (H_2_) in the presence of an iron catalyst and other oxide promoters such as K_2_O, Al_2_O, and CaO. The reaction runs at around 400–600 °C for efficient catalysis and up to 200–400 atmospheres of gas pressure to enhance entropy, an energy-hungry reaction that sucks up about 1% of global energy production and is thermodynamically exothermic [24].
N_2(g)_ + 3H_2(g)_ ⇌ 2NH_3(g)_ ∆H^o^ = −92 kJ(2)

While the nitrogen in the reaction above is extracted from the air, the hydrogen comes from natural gas (methane), oil, or coal through industrial processes that release CO_2_. Steam methane reforming is the most commonly used method to generate hydrogen, generating CO_2_ emissions. The current form of the Haber-Bosch process begins by generating hydrogen from fossil-fuel feedstocks, usually coal or oil. A reformer converts the feedstocks into a mixture of gases (syngas), which includes hydrogen. Thereafter, a carbon monoxide shift converter mixes water and the carbon monoxide from the preformed syngas to form carbon dioxide (CO_2_) and more hydrogen. The final steps involve the separation of hydrogen from ammonia synthesis by acid gas removal. At various steps of the process, CO_2_ is released (Figure 2). For every molecule of natural gas (methane) used, three molecules of CO_2_ are generated, and 1.6 tons of CO_2_ is emitted per ton of ammonia produced from the most efficient ammonia production plants [25,26].
CH_4_ + H_2_O + 4N_2_ ↝ 8NH_3_ + 3CO_2_
(3)

### 3.2. Blue Ammonia

Efforts by engineers across the world to make ammonia production less energy consuming and sustainable gave rise to the concept of blue ammonia. Blue ammonia, like brown ammonia, is produced from hydrocarbon feedstocks, but carbon capture utilization and storage (CCUS) technologies are integrated into ammonia production plants to sequester the resulting CO_2_. Of all the CCUS technologies known, amine absorption technology is the most widely used and commercially available [27]. Amines (or alkanolamines) are organic compounds with a basic nitrogen atom. They can be used to separate CO_2_ from the gas stream during ammonia production through the exothermic reaction of CO_2_ with an amine. Another CCUS technology is based on the principle that CO_2_ from any gas mixture (syngas) can be separated by cooling and condensation [28]. The technology, termed cryogenic separation, facilitates the direct production of liquid CO_2_, which can be transported. Although the amount of energy required for cooling in the process is relatively high and water must be removed to prevent cooling of the blocks by gas flow, the use of membranes in the gas separation process is promising [29]. Some of the membranes known to decompose CO_2_ are palladium membranes, polymeric membranes, and zeolites [30].

Another notable CCUS technology uses an adsorption device, rotary concentrator, on solids. The solid materials used for the adsorption include activated carbons, activated aluminum oxide (Al_2_O_3_), clays, zeolites, and silicon dioxide (SiO_2_). A modified version of this technology is pressurized swing adsorption (PSA), in which the gas mixture flows in the direction of the packed bed of the adsorbent at high pressure until the concentration of the desired gas to be separated reaches equilibrium [31]. The captured CO_2_ can be stored by several methods and used for a variety of production processes, including increased oil recovery, coal bed methane extraction, and deep ocean injection, among others [26,32]. In the long run, CCUS blue ammonia production technology will not be beneficial as high energy is still being used to drive the process, and the lack of CCUS infrastructure as well as transportation of CO_2_ poses yet another challenge.

### 3.3. Green Ammonia

The ammonia production process targeted at reducing or completely removing carbon dioxide emissions birthed the concept of green ammonia. To achieve zero carbon emissions during ammonia production, renewable feedstocks coupled with reduced energy usage are harnessed. At present, the most desirable but expensive green ammonia production method generates hydrogen from water electrolysis powered by solar, wind, hydroelectric, or geothermal energy [3]. This approach is also known as electrochemical ammonia synthesis (EAS). Electrolytes used for the EAS are diverse; they include solid electrolytes such as polymer electrolyte membrane (PEM) and anion exchange membrane (AEM), chlorine salts, melt hydroxides, and acidic electrolytes in liquid form [33]. Electrolysis in the latter is mostly done by the deposition of ammonium salts in solution to cause rapid changes in the pH of the solution. The low solubility of nitrogen often hinders electrolysis in its solutions. Thus, gas diffusion in electrodes is required for high efficiency and production rates [26].

In the EAS method, electrocatalysts commonly used based on the physical state and pH of the electrolyte include precious metals, metal nitrides, and metal oxides [34]. Transition metal-free catalysts such as black phosphorus and nitrogen-doped carbons are also known catalysts. The use of these catalysts minimizes the loss of nitrogen and improves process efficiency for high ammonia synthesis [35]. One of two reactors could be used to conduct the electrolysis: hydrogen generation reactors or nitrogen reduction reactors. The latter is preferred for low-temperature applications and gives more yield in downstream ammonia synthesis. Higher ammonia production efficiency can, however, be achieved in the hydrogen generation reactor by adding ZrO_2_ to the ruthenium catalyst. Similarly, reducing the number of protons on the catalyst surface by using high-pH electrolytes has also been shown to solve the underproduction problem in the hydrogen generation reactor [36]. The source of nitrogen is also crucial in the EAS method. The moisture content of the air used as the nitrogen source is an important parameter that affects the ammonia conversion rate. Using high-purity nitrogen from the air with reduced moisture will significantly increase ammonia synthesis [37].

Albeit the innovation of EAS to produce ammonia in an environmentally friendly manner, energy consumption is still unacceptably high due to the high current density utilized for hydrogen production, and the process occurs at a low capacity (10^−9^ to 10^−11^ mol cm^−2^ s^−1^) [38]. The water electrolyzer used in EAS requires a continuous supply of high-purity, pretreated water for its operation. Consequently, nine tons of water are consumed for every ton of hydrogen produced, and for the production of an amount of ammonia by EAS through water electrolysis, approximately double the amount of water is required, deepening the worldwide water crisis [3].

## 4. Biological Ammonia Production

Biological approaches are considered eco-friendly as they are natural processes that do not produce any harmful by-products. There are several approaches for biological ammonia production, including nitrogen fixation, nitrification, nitrate/nitrite reduction, urea hydrolysis, metabolic engineering of microorganisms, and in vitro ruminal microbial fermentation of protein biomass, but the most reported methods are biological nitrogen fixation (BNF) and metabolic engineering of microorganisms. Biological ammonia production by rumen bacteria fermentation of protein biomass, as experimented on in this review, is a relatively new approach and has shown the potential to complement ammonia bioproduction.

### 4.1. Biological Nitrogen Fixation by Nitrogenase

Biological nitrogen fixation (BNF) is a natural process that converts atmospheric molecular nitrogen (N_2_) to ammonia (NH_3_). BNF, an ATP-dependent reduction reaction catalyzed by the nitrogenase enzyme, is responsible for approximately half of the bioavailable nitrogen that supports all life forms [39]. Relative to the Haber-Bosch process, which requires high temperature and pressure conditions to break down molecular nitrogen, nitrogen-fixing microorganisms produce ammonia at ambient temperature and pressure. Nitrogen-fixing microbes are robust and have been explored to produce biofertilizers in commercial quantities [40,41]. Researchers are actively making attempts to mimic the natural process of BNF by isolating nitrogen-fixing bacteria (Figure 3) and nitrogenase for synthetic ammonia production. The major challenge with this research effort is that nitrogenase catalysis is highly energy dependent, making its reaction rate slower than most enzymes in nature [42].

The main microorganisms that possess nitrogenase and carry out nitrogen fixation are the genus *Rhizobia*, which colonizes the root of legumes, and species in the genera *Azotobacter* and *Klebsiella* that can fix nitrogen without parasitizing plant roots. The latter group is the main focus of research on synthetic BNF [43,44]. Nitrogenase requires up to eight molecules of ATP to produce a molecule of ammonia in an anoxic condition. Although the reaction mechanism of nitrogenase is unclear due to its multiple interrelated subunits, scientists have attempted to construct a heterologous expression system for *Klebsiella* nitrogenase subunits in *E. coli* [45]. Similarly, heterologous expression of the *Klebsiella* nitrogenase gene cluster has been constructed in *E. coli* and yeast to understand the mechanism by which nitrogenase functions without oxygen as well as to increase its activity [46,47]. Various studies have also investigated how nitrogen-fixing bacteria can function under aerobic conditions without inactivating nitrogenase. Such research involves the use of polysaccharide membranes to protect nitrogenase from oxygen exposure [47,48].

For industrial applications of nitrogen-fixing bacteria, some biotechnology companies have engineered *Enterobacter* sp. lacking glutamine due to low expression of the transcription factor *GlnR* to increase intracellular glutamine and, consequently, synthesize ammonia in the presence of nitrogenase [49]. Steady nutrient supply through BNF has also been successful with non-leguminous crop plants such as corn. The use of anaerobic microflora is also a known strategy for ammonia bioproduction by BNF, and a plethora of methodologies for ammonia recovery have been established. A notable one is the evaporation of solution following fermentation and pH increases [50,51,52].

### 4.2. Cell and Metabolic Engineering for Ammonia Production

Various biomasses, including food waste, microbial biomass, and protein-rich crop residues, can be fermented by engineered microorganisms whose metabolisms are well understood for ammonia bioproduction. In a metabolic engineering study on the conversion of protein wastes into biofuels and ammonia using microbes, the *codY* gene (a transcriptional regulator), in *Bacillus subtilis* was knocked out. The *codY* gene regulates the activity of several other genes involved in different processes, such as producing branched-chain amino acids (*ilvABHCD* and *leuABCD*), removing amino groups from other molecules (*ybgE*, *ald*, *yhdC*, *appBC*, and *dppBC*), and inhibiting the expression of genes that cause protein breakdown and uptake (*yhdG*, *appBC*, and *dppBC*). In bacteria, proteins are encoded for amino acid biosynthesis by the *ilv-leu* operon. The deletion of the *codY* gene removed regulatory constraints on this operon, causing a significant increase in the production and uptake of branched-chain amino acids (BCAA) due to the derepression of the *ilv-leu* operon and subsequent upregulation of genes responsible for BCAA synthesis.

In addition to the deletion of the *codY* gene, the *BkdB* gene in Bacillus subtilis was also knocked out. *BkdB* is a lipoamide acyltransferase enzyme that helps in the biosynthesis of branched-chain fatty acids by converting branched-chain keto acids into their acyl-CoA derivatives. This conversion inhibits the production of biofuels and ammonia. The *BkdB* gene knockout had a significant impact on the production of branched-chain fatty acids in Bacillus subtilis. Obstruction of production resulted in increased availability of metabolic precursors for the production of biofuels and ammonia. To completely transform *B. subtilis* to favor ammonia synthesis, an alcohol dehydrogenase gene, *leuDH,* and two-keto-acid decarboxylase were overexpressed. *LeuDH* is an alcohol dehydrogenase gene that plays an important role in the conversion of amino acids to alpha-keto acids, while two-keto-acid decarboxylase is an enzyme that catalyzes the decarboxylation of alpha-keto acids, which are important metabolic intermediates in amino acid biosynthesis. Overexpression of LeuDH increased the rate of amino-acid nitrogen reflux, which helped to increase the efficiency of protein conversion. Similarly, overexpressing two-keto-acid decarboxylase led to the increased availability of metabolic precursors such as alpha-ketoisocaproate (*KIC*) and alpha ketoglutarate (*AKG*) for the production of ammonia. The resulting final strain of *B. subtilis* was employed in the fermentation of protein biomass obtained from *E. coli* cells. This process produced ammonia with a theoretical yield of about 50% [53].

A similar study on ammonia production from amino acid-based biomass-like sources using engineered *E. coli* has been reported [54]. Since *E.coli* assimilates ammonia intracellularly [55], the two genes involved in the ammonia assimilation pathway, *glnA* and *gdhA* which are both glutamine assimilation genes, were knocked out to enhance ammonia production. *glnA* encodes for enzyme glutamine synthetase (GS) and catalyzes the conversion of glutamate and ammonia to glutamine, while *gdhA* encodes for the enzyme glutamate dehydrogenase (GDH) and catalyzes the reversible conversion of glutamate and ammonia to alpha-ketoglutarate. The deletion of *glnA* promotes the extracellular leaching of ammonia, while the deletion of *gdhA* increases ammonia flux to produce more glutamate, a known precursor of ammonia. In this study, deleting the two genes redirected the nitrogen assimilation pathways in *E. coli* toward ammonia production, resulting in a peak titer yield of 458 mg/L, equivalent to an overall yield of 47.8% [54].

Further studies on the metabolic engineering of *E. coli* for ammonia production converted different food wastes, including soy sauce cake, *mirin* cake, and tomato peel, to ammonia. Using metabolic profiling to assess the correlation between substances in the media (amino acids, sugars, and organic acids) and ammonia production, glucose was implicated as an inhibitor of ammonia production. When glucose was added to the amino acid-containing medium at different concentrations, a negative correlation with ammonia production was obtained. Thus, *E. coli* was engineered to hinder the inhibitory effect of glucose by knocking out the transporter gene, *ptsG*, and the phosphotransferase system, which transports glucose and other sugars. Briefly, the polymerase chain reaction (PCR) technique was used to amplify and copy specific fragments of genes that encoded resistance to *pts’G-Kim* and *glnA-Km* (amplified from pKD13) using primers *ptsGF* and *ptsGR.* The amplified DNA fragments were then transferred into *E. coli* cells through electroporation. Following the transfer, *E. coli* cells were grown on LB agar containing specific antibiotics—ampicillin and kanamycin. This allowed only the cells that had taken up the amplified DNA fragments to survive and grow, while the others died off. By repeating this process with different combinations of DNA fragments and antibiotics, more varieties of *E. coli* strains with different genetic modifications, such as *AptsG* and *AglnA,* were created. To ensure that the modified DNA fragments had been inserted into the correct location in the *E. coli* genome, PCR was used to amplify and sequence the insertion region using insertion-checking primers. The resulting *E. coli* strain succeeded in producing ammonia in a glucose-containing amino acid medium, with up to 73% yield [56]. In the studies described above, ammonia was, however, produced intracellularly. As a result, theproduced ammonia can still be used up by these microbes for growth [55]. Therefore, a system that can produce ammonia extracellularly without impeding microbial growth may improve productivity.

Studies on yeast for extracellular ammonia production have been attempted. Prominent among such studies is the use of yeast cell surface engineering (YCSE) systems to avoid ammonia toxicity and assimilation. In YCSE, the protein to be converted to ammonia is displayed on the cell surface, usually by the attachment of a secretory signal to the N-terminus of the target protein and a signal sequence, an α-agglutin containing a glycosylphosphatidylinositol anchor, on its C-terminus. Briefly, the plasmid for yeast cell surface display of L-amino acid oxidase was constructed by synthesizing and inserting the codon-optimized sequence of HcLAAO (L-amino acid oxidase) into pULDl, resulting in a plasmid named pULDl-HcLAAO. A strep-tag negative control plasmid called pULDl-s was also constructed. The yeast strain *Saccharomyces cerevisiae* BY4741/sedlA was utilized to display HcLAAO on the cell surface. The constructed plasmid was then introduced into the yeast strain. Yeast cells were then transformed, grown in a synthetic dextrose medium and cultured in SDC buffer at pH 7.0. Using this approach, up to10^6^ target proteins could be displayed on the yeast cell surface, which are then used as biocatalysts for enzyme immobilization [55,57,58].

Ammonia production from soybean residues has been successful with the YCSE technique [59]. Amino acid catabolic enzymes that produce ammonia from amino acid precursors, such as ammonia lyases, have attracted interest for their efficiency in being displayed on the yeast cell surface because their catalysis does not require cofactors, unlike nitrogenases. With yeast cells displaying glutamine ammonia-lyases, ammonia was produced from glutamine solution, reaching a titer of up to 3.34 g/L and an efficiency of 83.2% [59]. The limitation of this approach is that only glutamine, of the 20 amino acids, can be utilized. Interestingly, L-amino acid oxidase with a broad substrate specificity can be displayed for ammonia production from several amino acids [60,61]. These are lab-scale studies that may be difficult to transition to an industrial scale for eco-friendly biological ammonia production. Table 1 shows a summary of the metabolic engineering route for biological ammonia production.

### 4.3. Ammonia from Wastewater Treatment Plants

Microbial fuel cell technology can be used to produce ammonia in wastewater treatment plants through a process called ammonia oxidation [62,63]. Ammonia oxidation involves the use of specialized bacteria that are capable of oxidizing ammonia to produce electrons, which can then be used to generate electricity. In a typical microbial fuel cell system for ammonia production, the wastewater is first pumped into an anaerobic anode chamber. The anaerobic environment allows the bacteria to break down organic matter in the wastewater, releasing electrons in the process. The bacteria responsible for ammonia oxidation are then introduced into the anode chamber. These bacteria are able to use the electrons produced by the organic matter breakdown to oxidize ammonia in the wastewater [64]. Consequently, the wastewater is cleaned up, and ammonium is removed and converted into harmless gaseous N_2_ [65].

Ammonia can also be generated in wastewater treatment plants through ammonification. Ammonification, the breakdown of food waste, human waste, and other nitrogen-containing biological materials present in wastewater, converts the nitrogen-containing organic matter into ammonia [66,67]. Following the removal of large solids and debris from influent wastewater [68], the resulting wastewater is made to undergo a series of treatment processes to reduce nutrient levels, including a specialized approach called biological nutrient removal [69]. This process employs specific anaerobic bacteria such as *Clostridium perfringens*, *Peptostreptococcus*, *Actinomyces meyeri*, *Bifidobacterium species*, *Propionibacterium*, *Bacterioides*, and *Fusobacterium* [70] to break down organic matter and convert nitrogen compounds into ammonia. The ammonia produced is then further transformed into nitrate and nitrite ions through a process called nitrification [71,72]. Although there have been numerous studies conducted on ammonia production from wastewater treatment plants [65,73,74,75,76,77,78,79,80,81,82], most of the processes involved are highly energy-intensive and economically non-viable [65].

### 4.4. Hyper Ammonia-Producing Bacteria Route

The digestive compartment of ruminant animals, the rumen, is a biorefinery for ammonia production. Ruminal microorganisms can break down plant materials containing carbohydrates and proteins in their feeds for energy. The products of protein degradation, including peptides and amino acids, are metabolized to protein and/or ammonia. The microbial protein thus formed is required for animal products, but the ammonia is absorbed from the rumen, metabolized, and excreted in the urine. This is an inefficient use of dietary proteins with devastating consequences for the environment through environmental nitrogen pollution [83].

Several studies in the animal sciences have sought strategies to promote microbial protein synthesis and regulate ammonia production. These studies revealed the identity of a certain group of bacteria whose rate of ammonia production is much higher than can be used up by the ruminal microbes for other functions, including microbial protein synthesis [84,85,86]. This group of bacteria, known as the hyper-ammonia-producing bacteria (HAB), can effectively convert dietary protein to surplus ammonia [87,88]. This type of natural ammonia is produced when the digestive systems of humans and animals undergo a biochemical reaction leading to the breakdown of nitrogen-containing amines (NH_2_) in proteins into ammonia or the ionic form (ammonium). It is referred to as biological ammonia.

The first step towards the degradation of amino acids is deamination, which is the removal of an amine group to convert it to ammonia. It has been reported that amino acid deamination in the rumen produces more ammonia than can be utilized by the bacteria [89]. Deamination may occur through oxidation, reduction, hydrolysis, or the removal of elements. It helps to free the carbon skeleton by removing the amine group from the amino acid. Furthermore, deamination could be carried out on either a single amino acid, pairs of amino acids as in the case of the Stickland reaction, or a combination of amino acids and a non-nitrogenous compound, all resulting in ammonia and keto-acids as major products [90].

The next biochemical reaction is called ammonification, which is the second stage of mineralization [91]. Useful energy can also be derived metabolically by bacteria and related microorganisms through ammonification. Ammonium (NH4+) is thus produced by microorganisms, and if in excess, it is excreted into the environment as nutrients for uptake by plants or as feedstock for further nitrification [91]. HABs have been implicated in converting ~50% of ruminal dietary protein to ammonia [92,93,94].

HABs are found in cattle rumen or swine manure stored in the pit [95,96,97]. Additionally, HABs thrive in the rumen of the hay-fed cattle compared to grain-fed cattle [84] because the pH of hay-fed cattle rumen environment is relatively neutral, thus providing a favorable condition for their growth compared to the slightly acidic pH (<6.0) observed in grain-fed cattle [98]. HABs are capable of producing up to 40 mM (0.6812 mg/L) of ammonia in peptone-amino acid medium, depending on energy and carbon source [96,99]. HABs can operate in both anaerobic and aerobic environments, but anaerobic-HABs are more prominent and of major concern because they convert a large percentage of dietary protein in the rumen to ammonia [100]. Although HABs are detrimental to ruminant metabolism due to excess ammonia generation causing toxicity to rumen microbes and hyperammonemia in farm animals [100], they can be harnessed as a sustainable source for large-scale ammonia production with low energy requirements and zero emissions.

There are several strains of hyper-ammonia-producing bacteria (HAB) with different biological ammonia-production capacities. *Selenomonas ruminantium*, *Peptostreptococcus elsdenii*, and *Bacteroides ruminicola* are HAB strains that are capable of producing at least 1 µM of biological ammonia on a lab scale through deamination. *S. ruminantium* catabolizes cysteine hydrolysate, while *P. elsdenii* breaks down casein hydrolysate and specific amino acids (L-serine, L-threonine, and L-cysteine) to produce biological ammonia [101,102]. Depending on HAB strain and environmental conditions, it is also possible to produce much higher concentrations of biological ammonia (>24 mM) [99].

## 5. Biomanufacturing

### 5.1. Conceptual Bioprocess Flow

There have been a few attempts to employ HAB in the fermentation of dietary proteins to produce biological ammonia. For instance, in a study to investigate the affinity of hyper-ammonia-producing bacteria (HABs) to produce biological ammonia, three hyper-ammonia-producing ruminant bacteria (*Clostridium aminophilum, Peptostreptococcus anaerobius, and Clostridium sticklandii*) were anaerobically cultured using five different organic nitrogen substrates: soy protein isolate (SPI), blood meal (BM), feather meal (FM), dried fish meal (DFM), and yeast extract (YE). The study examined the affinity of these HABs to produce biological ammonia. The ability of these bacteria to produce ammonia when grown in pure culture with various protein sources was then assessed to determine which bacteria species and protein substrate produced the highest concentrations of ammonia. Results showed that all three bacteria produced ammonia at various rates, depending on the organic nitrogen substrate used. *Clostridium aminophilum* and SPI produced the highest biological ammonia concentration of about 7.23 mM [103]. With this finding, the fermentation of proteins using HABs could be harnessed for sustainable biological ammonia production.

As earlier stated, there have been few studies on the fermentation of hydrolyzed proteins, peptides, and amino acids. HABs synthesize varying amounts of biological ammonia from different peptides and amino acids [95,103]. In fact, before fermentation, HABs inherently secrete proteases to initially hydrolyze proteins [95]. The protein hydrolysates, which comprise oligopeptides, peptides, and amino acids, are easily absorbed by HABs [95]. However, there has not been any study that compares hydrolyzed and unhydrolyzed proteins during biological ammonia production. Therefore, to understand the importance of pre-fermentation hydrolysis, future studies should investigate the effect of pre-fermentation enzymatic hydrolysis of proteins on biological ammonia production. Hence, we include protein hydrolysis as an important step in the conceptual bioprocess flow for biological ammonia biomanufacturing (Figure 4).

### 5.2. Protein Hydrolysis

Proteins are large polymers of amino acids joined together, primarily, by peptide bonds. Polypeptides and peptides are long and short chains of amino acids, respectively. Most of the proteins present in soybeans and soybean meals are in the form of storage globulins [104]. The polypeptide chains of the proteins are entangled into a three-dimensional complex structure by several hydrogen and disulfide bonds, amounting to a molecular weight of up to 600,000 kDa. Soybean proteins are insoluble in water at their isoelectric point. The two major types of proteins in soybean seeds are glycinin and conglycinin, which form ~80% of the total soybean protein. Glycinin and conglycinin have similar secondary structures: 57% random coils, 38% beta-sheet, and 6% alpha-helix. In glycinin, glycine, tyrosine, and tryptophan are wrapped inside the globular structure, whereas in conglycinin, tryptophan is exposed. Additionally, glycinin has two major sub-proteins: acidic glycinin of ~40 kDa and basic glycinin of 20 kDa, whereas conglycinin has three sub-proteins: alpha-conglycinin, beta-conglycinin, and gamma-conglycinin with 68 kDa, 175 kDa, and up to 200 kDa, respectively [105]. These protein polymers must be deconstructed and utilized by microbes for ammonia bioproduction. Products of protein hydrolysis—amino acids and peptides—are referred to as protein hydrolysates. Relative to intact proteins, protein hydrolysates have increased solubility, which could enhance their bioavailability. Complete protein degradation generates amino acids that are then further catabolized and deaminated to release ammonia. Peptides, products of partial protein degradation with short chains of amino acids, are also utilized by ruminal microbes for ammonia biosynthesis. The catabolism of peptides by ruminal microbes involves the activity of peptidases secreted by these microbes. Peptidases abound in the rumen and have different substrate specificities [106]. However, several studies have shown that external enzymes can be highly effective in enhancing soybean protein degradation and utilization of the resulting products—short peptides and amino acids—in the rumen [107,108]. In vitro digestion of proteins for various applications has been extensively studied [109,110,111,112,113,114,115,116]. The most common methods for protein hydrolysis are biological (enzymatic) and thermochemical hydrolysis.

#### 5.2.1. Biological/Enzymatic Protein Hydrolysis

Enzymatic hydrolysis is a safe and effective approach to processing proteins into hydrolysates with improved functional properties. Enzymes used for protein hydrolysis are also called proteases. The operational conditions for enzymatic hydrolysis are environmentally friendly and reduce the formation of by-products [117]. Common enzymes used include pepsin, trypsin, chymotrypsin, papain, and various extracted fungal and bacterial proteases such as flavourzyme, alcalase, neutrase, and protamex. The form of soybean proteins (toasted or untoasted soybean meal (SM) protein, processed SM, heated SM, defatted SM, native SPI, etc.) and the enzymes/proteases impact the degree of hydrolysis (DH) and functional properties of the resulting hydrolysates [118]. The extent of protein denaturation that gives rise to different protein forms determines the level of resistance of the protein to proteolysis. In toasted SM, for instance, beta-conglycinin is effectively hydrolyzed by many different endo-proteases [119]. On the other hand, higher resistance to proteolysis was observed with beta-conglycinin than with glycinin by rumen bacteria [120]. These results informed the need to explore the potential of using multiple enzymes together or sequentially to achieve a complete breakdown of any protein form.

#### 5.2.2. Multi-Enzymatic Hydrolysis

Multi-enzymatic hydrolysis of proteins is the combination of different enzymes to digest protein molecules into peptides and amino acids. Based on their catalytic mechanisms and amino acid sequence, proteases are diverse and have different activity and specificity. While serine proteases use serine residue in their active sites to cleave peptide bonds, cysteine proteases, and metalloproteases use cysteine residue and metal ions to cleave peptide bonds, respectively. Some proteases, such as trypsin, cleave specific peptide bonds only (after lysine and arginine residues); yet other proteases have broader specificity, especially the industrial proteases. Proteases also differ in their sites of catalytic action. They could either cut from within (endoproteases) or from the terminal amino acid residues (exopeptidases). Industrial proteases are usually a mixture of different enzymes purified from bacteria or fungi. Thus, a single industrial enzyme could be both an endo- and exoprotease. For example, flavourzyme is a mixture of proteases with both endo- and exopeptidase activities produced and purified from Aspergillus oryzae [121].

Studies have assessed the potential of different industrial proteases in simultaneous or sequential combinations for the hydrolysis of different protein isolates. These studies revealed that proteases used in combination increase the functional properties of the resulting protein hydrolysates [109,122,123]. The concentrations of the constituent enzymes in combination have also been shown to increase the DH of protein isolates. In a study, DH was shown to increase significantly when a higher concentration of alcalase in an alcalse-flavourzyme combination or an alcalse-corolase combination was used to hydrolyze potato pulp protein [124]. The alcalase-flavourzyme combination gave the highest DH in both the hydrolysis of potato pulp protein and poultry meal [122,124]. The protease combination for hydrolysis in any bioconversion is process specific. Several studies have revealed the different combinations that work for specific bioconversion processes [123,125,126]. There is a paucity of data on protein hydrolysis for biological ammonia production. Thus, research efforts in this area would enhance the development of optimal parameters for biological ammonia production.

### 5.3. Leading HABs for Biological Ammonia Production

*Clostridium aminophilum* (amino and philos meaning loving amino acids) is a species of bacteria in the family *Clostridiaceae*. It is an atypical gram-positive bacterium with a length and width of 1.0 µm and 1.5 µm, respectively. The cellular spores are non-motile and can withstand a temperature as high as 80 °C for 10 min. Classified as obligately anaerobic with an optimum growth temperature of 25 °C to 45 °C, they utilize glutamine, glutamate, serine, and histidine as carbon sources [127]. Their end products of fermentation include ammonia, acetate, and butyrate, along with tinctures of lactate and succinate.

*Clostridium sticklandii* is also a species of bacteria in the genus *Acetoanaerobium* and the family *Peptostreptococcaceae*. It is also a gram-positive obligate bacterium that utilizes threonine, arginine, serine, threonine, cysteine, proline, and glycine as preferable carbon sources [128], in their research on the biology of bacteria, concluded that *C. sticklandii* can utilize pairs of amino acids through the Stickland reaction to bring forth fermentation products such as acetate, butyrate, and ammonia.

*Peptostreptococcus anaerobius*, unlike *Clostridium aminophilum*, is a non-spore-forming obligate bacterium. It is, however, gram-positive [129]. Their common carbon sources include leucine, serine, threonine, glycine, and phenylalanine [130]. *P. anaerobius* can also break down amino acids and peptones into ammonia, acetic, butyric, and isobutyric acids. It is, however, worthy of note that there is very little or no information in the literature on the genetic makeup and characteristics of HAB.

### 5.4. Factors Affecting Biological Ammonia Production

Several bacteria species have over time been identified with a speedy rate of ammonia production [131]. However, the ability of these bacteria to produce ammonia at an optimum rate is dependent on several factors. Research on the isolation and characterization of these bacteria has been carried out extensively to understand the factors responsible for ammonia production.

#### 5.4.1. Effect of Diet, Substrate, and Substrate Combination

According to [132], the rate at which ammonia is produced changes based on diet. The rate at which organic ammonia proceeds is also dependent on the substrate, its concentration, and the method of ammonia production utilized [132]. Isolation of bacteria by [130] identified *Peptostreptococcus* and *Clostridium* as ammonia-producing bacteria when soybean meal was a major ingredient in ruminant feed. Additionally, in a study to appraise the rate of ammonia production by HAB, pure cultures of *P. anaerobius*, *C. sticklandii*, and *C. aminophilum* were grown in vitro utilizing different substrates and in different combinations. Five different diets rich in organic nitrogen sources (soy protein isolate (SP), blood meal (BM), feather meal (FM), dried fish (DF), and yeast extract (YE), were utilized to discover the best protein substrate and bacteria species yielding maximal quantities of ammonia. Results showed that with the combination of all substrates, *P. anaerobius* produced the least amount of ammonia, followed *by C. sticklandii* and *then C. aminophilum*. The same quantity of organic ammonia was produced for combinations of BM alone, BM and YE, SP and YE, FM and YE, DF alone, DF and YE, and YE alone. The results recorded are in alignment with [127], where *C. aminophilum* was best in the production of organic ammonia. Moreover, *C. aminophilum* yielded the highest organic ammonia concentration (7.23 mM) when cultured on soy protein isolate alone [103]. The substrates were particularly used due to their richness in certain specific amino acids that aid the rapid multiplication of HAB species [103]. A careful study of the amino acid profile of the best experimental substrate (soy protein isolate) used by [103] revealed the presence of nine standard and essential amino acids, which can only be synthesized from food sources. They include histidine, isoleucine, methionine, phenylalanine, lysine, valine, threonine, leucine, and tryptophan [14,133]. Therefore, they concluded that the effect of substrate on HAB species was highly substantial to the extent that it accounted for about 17.0% of the variation, indicating that all bacteria responded differently to the substrate.

Moreover, following the growth of swine manure isolates and ruminal HAB on different sources of amino acids, there were pronounced variations in the growth and ammonia production of HAB [96]. As described in a previous study, HAB thrived well on tryptone, casamino acids, and a mixture of both, producing ammonia in high concentrations [99,130,131].

Similarly, in a study of ruminal ammonia-producing bacterial species, three experimental cows (A, B, and C)were put on dissimilar diets of timothy hay, grain mixture, and alfalfa hay, respectively [134]. Animals were later fed the feed mixture in varying proportions. Following the collection of rumen samples, bacteria strains were isolated and subsequently labeled as A, B, and C based on the source (animal) from which the samples were collected. Results showed a difference in the concentration of ammonia produced. A variation in the amount of organic ammonia produced has been attributed to diet. In addition, ref. [84] conducted research aimed at calculating and equating the ammonia production of mixed ruminal bacteria with their deamination rates and that of carbohydrate-fermenting ruminal bacteria, thus evolving a mathematical model of ammonia production and figuring out the population of HAB in the rumen of cattle fed predominantly with hay or grain-based diets. Results showed that the rate of ammonia production from mixed ruminal bacteria in cattle fed with hay doubled that of cattle fed predominantly with grains. In furtherance of this research, a mathematical model supporting the results was developed. The model showed that the hyper-ammonia-producing ruminal bacteria of cattle fed with hay were four times higher, thus indicating that the HAB of cattle-fed hay had a high maximal velocity of ammonia production.

#### 5.4.2. Effect of pH

Microorganisms are generally sensitive to their ambient hydrogen ion concentration (pH). Most ruminal bacteria are anaerobes and thus act optimally at neutral pH (pH range of 6.5–7.5). In the research to determine ammonia production from HAB, a pH of 7.0 was utilized [84]. Likewise, in research on ammonia production by ruminal microorganisms, ammonia production was determined utilizing filtered ruminal fluid from sheep using the method proposed by [135]. Substrates such as casein, trypticase peptone, soluble soybean protein, and soya peptone were utilized. The pH of the resulting liquid (supernatant) was brought to 7.0 by neutralization with KOH, and the rates of ammonia production were determined through linear regression. In addition, all incubations were also carried out at a pH of 6.7 to 7.0 in a similar study on ammonia production by monensin-sensitive ammonia producing bacteria [99]. However, other studies have indicated that there might be no direct relationship between rumen pH and the percentage of HAB organisms present in the ruminants sampled [131], suggesting that pH may not have any effect on the rate of ammonia production.

#### 5.4.3. Effect of Temperature

Temperature plays a very significant role in ammonia production by HAB. Different bacteria have different temperature ranges for optimal performance. In research on the isolation and identification of HAB from swine manure storage pits, all bacterial isolates were grown at 37 °C [96]. Likewise, the measurement of ammonia produced in ruminal fluid in vitro involving the dissolution of soluble soybean protein in water was also done at room temperature [132].

Furthermore, in research on the estimations of hyper-ammonia-producing ruminal bacteria by [84], ruminal bacteria were anaerobically grown at a temperature of 39 °C in a basal medium. Isolated ruminal hyper-ammonia-producing bacteria have also been previously cultured in a growth medium at 39 °C [136]. Likewise, in a study on the characterization of ruminal microorganisms responsible for ammonia production, sheep-filtered ruminal fluid was grown in vitro at a temperature of 39 °C [132]. These results show consistency in temperature values and thus suggest that HAB are generally mesophilic, acting optimally at temperatures between 20 and 45 °C.

#### 5.4.4. Effect of Time

The effect of time on the production of organic ammonia by HAB cannot be overemphasized. In the experiment on the potential of HAB to produce organic ammonia, results indicated that *C. aminophilum* yielded the most organic ammonia when cultured on soy protein isolate substrate for 96 h [103]. The main effect of time on organic ammonia production accounted for about 9.7% of the variation. It was, however, observed that *P. anaerobius* had very low growth rates, produced the lowest organic ammonia concentrations, and was unaffected by time [134] in their study of bacteria species from the rumen, they found out that the incubation period for ammonia production gradually increased from 72 h. In the long run, a 96 h standard time was chosen.

## 6. Conclusions

In this review, we have presented numerous attempts towards achieving sustainable, CO_2_ emission-free biological ammonia production, and although several promising approaches have been presented, their respective limitations have also been acknowledged. In addition to these renewable alternatives, we have proposed a novel strategy for generating biological ammonia through the fermentation of dietary proteins via the rumen hyper-ammonia-producing bacteria pathway. We recommend further exploration and investigation of this approach for adapting rumen microbial fermentation to laboratory-scale studies, with the ultimate goal of advancing this approach toward commercialization.

## Figures and Tables

**Figure 1 biotech-12-00041-f001:**
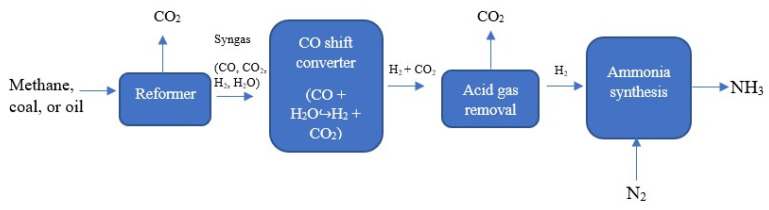
Haber-Bosch process flow chat.

**Figure 2 biotech-12-00041-f002:**
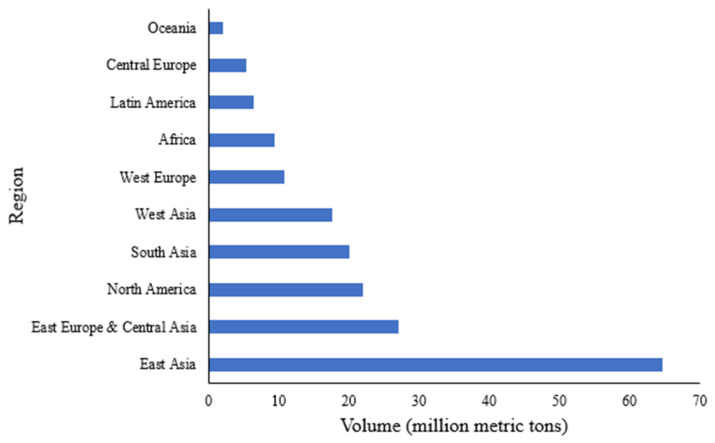
Regional Ammonia Production Capacities in 2020.

**Figure 3 biotech-12-00041-f003:**
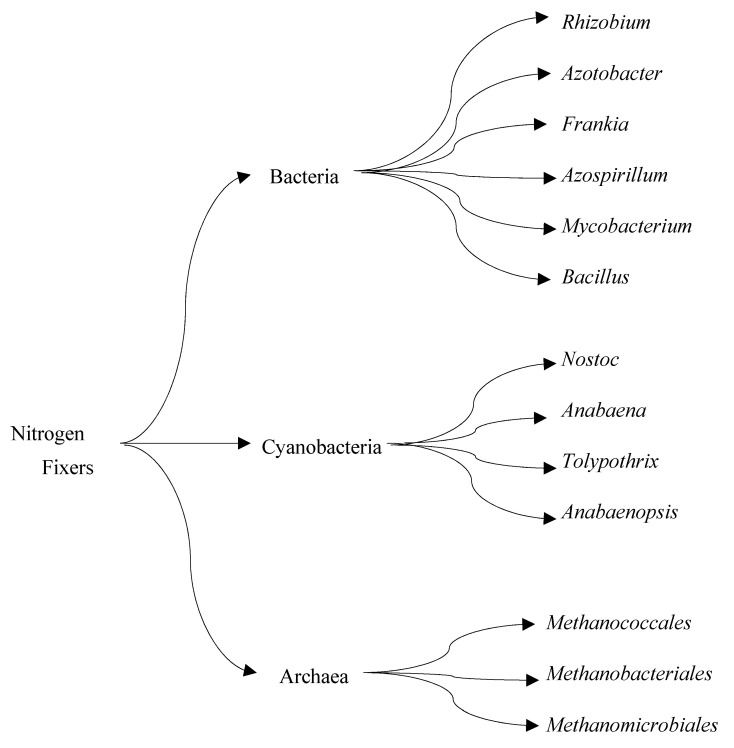
Three groups of nitrogen-fixing microorganisms.

**Figure 4 biotech-12-00041-f004:**
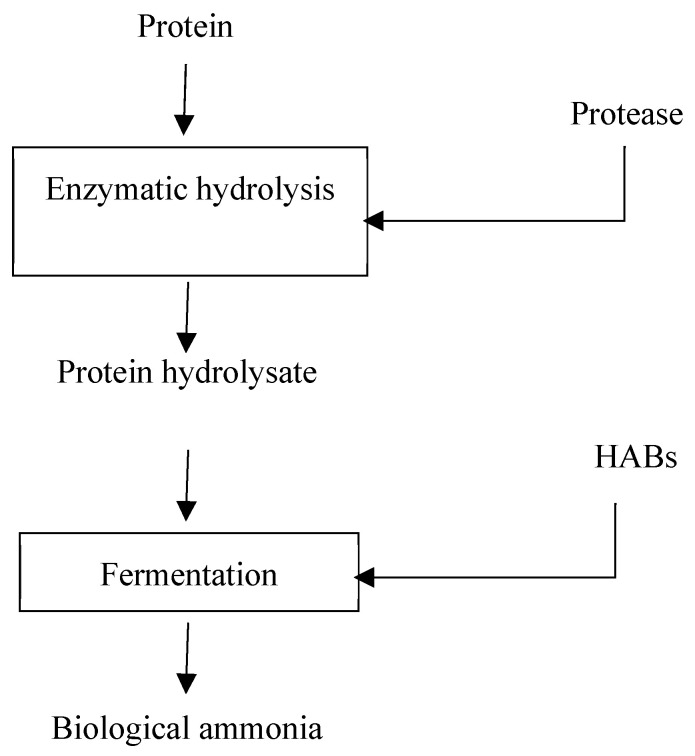
Conceptual bioprocess flow for biological ammonia production.

**Table 1 biotech-12-00041-t001:** Summary of metabolic engineering approaches for biological ammonia production.

Approach	Description	Host	Substrates	Ref
Gene knockout	Deletion of *CodY* gene which regulates genes: *ilvABHCD* and *leuABCD* (that produces branched-chain amino acids) ybgE, ald, yhdC, appBC and dppBC (which causes deamination) yhdG, appBC, and dppBC (which inhibits the expression of genes that cause proteolysis and protein uptake	*Bacillus subtilis*	Amino acid	[53]
Gene knockout	Deletion of gene *BkdB* which helps in the biosynthesis of branched chain fatty acids	*Bacillus subtilis*	Amino acid	[53]
Gene overexpression	Over expression of proteins *leuDH,* and two-keto-acid decarboxylase which respectively converts amino acids to important metabolic intermediates and increases the availability of metabolic precursors for ammonia production	*Bacillus subtilis*	Amino acid	[53]
Gene knockout	Deletion of genes glnA and gdhA which aids ammonia assimilation	*Eschericia coli*	Amino acid	[54]
Gene knockout	Deletion of *ptsG* (glucose transporter gene) and deletion of phosphoenol pyruvate (glucose transporter)	*Eschericia coli*	Soybean residue and food waste	[56]
Cell surface engineering	HcLAAO (L-amino acid oxidase) display on yeast cell surface by gene insertion	Yeast cells	Amino acids from soybean residue	[55]
Cell surface engineering	Glutaminase gene (*Ybas)* display on yeast cell surface by gene insertion.	Yeast cells	Soybean residue and glutamine	[59].

## Data Availability

Not applicable.

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
