# Peer review of "Trends in Biological Ammonia Production"

_biotech, 2023, doi:10.3390/biotech12020041_

Round 1

Reviewer 1 Report

- What is the link between the production of ammonia and political instability?

- It is a review article and therefor it is necessary to discuss the different applications of ammonia.

- When you talk about ammonia as a potentiel fuel, what do you think of the energy cost of its production compared to its energy value?

- You must add in your article the  ammonia obtained in certain wastewater treatment plans. It is an ecological source and not expensive.

- Line 253: write E coli i italic.

- Paragraph 5.2.1: you need to talk about the use of enzyme immobilization to have abetter production yield.

- Conclusion: number 6 not 5.

Reviewer 2 Report

The paper should present a review of the biological processes employed for ammonia production. I appreciated the effort made by the authors in collecting and reviewing papers on the subject but I’m sorry to say that the materials is not well-organized and the paper is not mature to deserve publication. Here the main drawbacks I found:

Abstract and conclusions are too general, the reading of them both do not allow to catch the nature and the value of the paper;

Information reported in sections 4 and 5 (that is the hearth of the paper) are not well-organized. It is very difficult to find the reasoning. It is impossible, if you do not have a strong biological knowledge. The authors must make a great effort to come to a mature, well-balanced story;

Table 1 and figures 3 and 4 are not of help to elucidate the content of the section; I suggest to produce more informative and attractive materials; with specific reference to Figure 3, are not cyanobacteria and archea bacteria as well?

Not clear to me the technology readiness level of the presented processes. Were they only tested at a lab scale? Are they ready to be used at industrial scale? If yes, at which cost? For that I think that the title is a bit misleading.

Finally, English is not my first language, but the quality of the language and sentences’ structure must be improved.

Finally, English is not my first language, but the quality of the language and sentences’ structure must be improved.

Reviewer 3 Report

I revised the paper intitled: “Trends in Biological Ammonia Production.

This is an interesting study, with plenty of valuable insights.

Formatting is needs, as the authors must follow exactly the formatting indicated in the authors' instructions, and some aspects are not correct – for example, line spacing, in lines 61-68, is not correct.

 In all pictures and schemes the resolution of the letters and text is very low, and this should be reviewed.

The references used could possibly be more recent, I only found 1 publication cited from 2023, and 5 from 2022, the rest being from earlier dates.

Additionally, other revisions suggested:

Title:

Line 9: CO2 instead of CO2

Lines 9-10: I suggest the authors to revise the sentence “To mitigate this challenge, many research 9 efforts have developed bioprocessing technologies to make biological ammonia” as the meaning is not clear. I think that maybe the authors meant: “To mitigate this challenge, many research efforts have been made, to develop bioprocessing technologies to make biological ammonia.”

Introduction:

Lines 20, 21: I suggest the author to replace “NH3” by “NH3

Line 25: “Anhydrous ammonia can be a gas, a liquid or a solid”

Line 31: NH4OH

Line 31: “Gaseous ammonia” or “ammonia gas”?

Lines 44-46: Maybe: “Concerns have been growing about their sustainability and, more importantly, the effects of their energy intensity level on climate change and the environment.”?

Line 47: CO2 instead of CO2

Line 49: N2O instead of N2O

Lines 70-72: “Ammonia is a versatile product, used in the production of fertilizers, refrigerant gas, water purification and other industrial applications such as the manufacture of explosives, textiles, pesticides, plastics and dyes.”

Line 229: Please remove the word “methods”, as it is repeated in the same sentence.

Line 252: Klebsiella

Line 253: E. coli

Figure 3 is truncated on the left side and not italicized

Line 304: Enterobacter

Line 326: Bacillus subtilis

Line 345: E.coli – replace by E. coli

Line 605: Acetoanaerobium

Line 608: C. sticklandii

Line 613: P. anaerobius

Line 680 and 682: 7.0 (not 7)

Line 695 and 697: Correct 39oC

Line 702: Please remove 5.4.4., if there is no data, it makes no sense

Minor changes needed - please see comments above

Reviewer 4 Report

Comments on Manuscript ID biotech-2378755

Title                 Trends in Biological Ammonia Production

Authors          Adewale Adeniyi et al

This review article is well informative about the Biological Ammonia Production dynamics and needs to address the following concerns

Comments

1.    In the text, this manuscript should be considered as a review article.

2.    The political instability term needs to be removed from the 9th line of the abstract.

3.    Paragraph 2.1 under title scale of production is irrelevant in the review article and should be removed from the manuscript.

4.    In section 2.2, the company name of Manufacturer BASF has been mentioned twice, which needs to be deleted from the text.

5.    From section 4.1, paragraph 2, the author needs to remove the name of the company and should keep their reference as the citation.

6.    In the second paragraph of section 4.2, in the last sentence, the author stated about the increases of pH which needs to provide the rand of pH increase by describing whether this increment is under permissible limit or crossing the limits in toxicity.

7.    The conclusion should be number 6 of the text.
